# Secrecy Performance Analysis of Cooperative Multihop Transmission for WSNs under Eavesdropping Attacks [note 1]

**DOI:** 10.3390/s23177653

**Published:** 2023-09-04

**Authors:** Yosefine Triwidyastuti, Ridho Hendra Yoga Perdana, Kyusung Shim, Beongku An

**Affiliations:** 1Department of Software and Communications Engineering in Graduate School, Hongik University, Sejong City 30016, Republic of Korea; yosefine@mail.hongik.ac.kr (Y.T.); hendra@mail.hongik.ac.kr (R.H.Y.P.); 2School of Computer Engineering and Applied Mathematics, Hankyong National University, Anseong City 17579, Republic of Korea; kyusung.shim@hknu.ac.kr; 3Department of Software and Communications Engineering, Hongik University, Sejong City 30016, Republic of Korea

**Keywords:** cooperative transmission, eavesdropper, multihop relay, node selection, physical layer security, secrecy outage probability

## Abstract

Multihop transmission is one of the important techniques to overcome the transmission coverage of each node in wireless sensor networks (WSNs). However, multihop transmission has a security issue due to the nature of a wireless medium. Additionally, the eavesdropper also attempts to interrupt the legitimate users’ transmission. Thus, in this paper, we study the secrecy performance of a multihop transmission under various eavesdropping attacks for WSNs. To improve the secrecy performance, we propose two node selection schemes in each cluster, namely, minimum node selection (MNS) and optimal node selection (ONS) schemes. To exploit the impact of the network parameters on the secrecy performance, we derive the closed-form expression of the secrecy outage probability (SOP) under different eavesdropping attacks. From the numerical results, the ONS scheme shows the most robust secrecy performance compared with the other schemes. However, the ONS scheme requires a lot of channel information to select the node in each cluster and transmit information. On the other side, the MNS scheme can reduce the amount of channel information compared with the ONS scheme, while the MNS scheme still provides secure transmission. In addition, the impact of the network parameters on the secrecy performance is also insightfully discussed in this paper. Moreover, we evaluate the trade-off of the proposed schemes between secrecy performance and computational complexity.

## 1. Introduction

The main focus of the next-generation network is on human-to-machine interactions and real-time communication by utilizing various tactile/haptic sensors and actuators that have a massive number with a small size but limited energy [1]. In 5G networks, the number of devices in a network can reach 1 million devices per square kilometer. Moreover, the connection density in 6G systems will reach $10^{7}$ devices/km${}^{2}$ [2]. However, a high density network is still prone to blockages when the destination is located in a far distance from the low-energy source, and there is no traffic distribution [3].

By using cooperative transmission, nodes in the network can be designated as relays that decode the received information from the source and retransmit the received information to the destination. Relay gives benefits to a communication network by extending its coverage, increasing the signal reception while reducing its energy consumption [4]. Due to these advantages, cooperative transmission has been implemented in various wireless systems [5,6,7]. Specifically, sensor nodes in wireless sensor networks (WSNs) collect confidential and sensitive information to a gateway or server through a cooperative multihop communication [8].

In practice, WSN with cooperative transmission is widely implemented in various industries, such as disaster mitigation, vehicular network, and battlefield. Security has become the main issue for the large-scale application of WSNs [9]. Due to the broadcast nature of wireless nodes, illegitimate users can easily wiretap data transmission. An eavesdropper can disconnect the transmission between sensor nodes or send a wrong message and cause an error. In multihop WSN where intermediate nodes directly access the message and retransmit it to the next hop, the end-to-end security for confidential messages is difficult to achieve if there is no countermeasure in the data transmission [10].

A higher number of nodes that relay confidential information can surely reach the destination in a more remote area, but it is also more susceptible to eavesdropping attack [11]. An illegitimate node that is located within the relays’ wireless range can overhear the broadcast messages [12]. The illegitimate node that only overhears the main channel transmission is known as a passive eavesdropper. On the other side, when the illegitimate node simultaneously transmits a jamming signal while overhearing the main channel transmission, it is known as an active eavesdropper and more advanced than a passive eavesdropper [13].

There are several considerable techniques to overcome an eavesdropper. The well-known technique is using data encryption in the network’s application layer. However, encryption needs careful key management and distribution in the source node and destination node. Encryption can also be deciphered by brute-force computing in an advanced eavesdropper node [14]. The new emerging technique for overcoming an eavesdropper is physical layer security (PLS), which exploits communications’ medium information to either enhance the main channel capacity, reduce the eavesdropper capacity, or both [15].

In this paper, we exploit the impact of secrecy performance on various eavesdropping attacks in multihop transmission. To enhance the secrecy performance, we propose two node selection schemes. The first node selection scheme can select the node in each cluster to minimize the eavesdropper channel. The second node selection scheme can select the node in each cluster to maximize the secrecy capacity. The main contributions of this paper can be summarized as follows:We exploit the impact of different eavesdropping attacks on the secure multihop transmission for WSNs. More specifically, in the passive eavesdropping attack, the eavesdropper only overhears legitimate users’ transmission. Different from the passive eavesdropping attack, the active eavesdropping attack can overhear each hop transmission and radiate the jamming signal to reduce the main channel condition at the same time. This scenario has not been studied in this literature.To enhance the secrecy performance, we propose two node selection schemes. The first scheme can select the node that minimizes the eavesdropper channel gains, called the minimal node selection (MNS) scheme. The second scheme, namely, the optimal node selection (ONS) scheme, can maximize the secrecy capacity of each cluster. We also consider the random node selection (RNS) scheme, which randomly selects the node in each cluster as a benchmark to compare the secrecy performance with the proposed schemes in a multicluster network.In order to find the relation between the system parameters and the secrecy performance, we derive a closed-form expression of the secrecy outage probability (SOP) with different eavesdropping attacks and the proposed node selection schemes. Specifically, we obtain the end-to-end SOP as the function of the number of clusters, number of nodes, target secrecy rate, main channel transmit SNR, and eavesdropper jamming SNR.From the numerical results, the active eavesdropper seriously affects secrecy performance compared with that of a passive eavesdropping attack. Additionally, ONS outperforms RNS and MNS secrecy performance in terms of SOP, while ONS requires a huge amount of channel information compared with that of other schemes.

The rest of this paper is organized as follows: Section 2 exploits previous works that motivate this research. Section 3 describes the system model of the proposed multihop relaying network, along with the passive and active scenarios of an eavesdropper attack and three node selection schemes. Section 4 analyzes the closed-form function of the system’s SOP for all cases as the combination of eavesdropper scenarios and node selection schemes. Section 5 presents the numerical results obtained from the derived analysis and simulations. Finally, Section 6 concludes the paper.

## 2. Related Works

Studies on PLS mostly focus on three main strategies: optimized resource allocation, secure beamforming/precoding, and antenna/node selection [16]. Duo et al. proposed joint UAV trajectory and power control optimization for securing UAV communications [17]. However, finding an optimal resource allocation in a wireless network is a complicated task that needs a strategic game to formulate the interactions between all nodes. Wang et al. obtained a Stackelberg equilibrium in multiantenna cellular networks through an iterative algorithm [18]. Luo and Yang in [19] formulated the cooperation between cellular user, D2D user, and active eavesdropper as a secrecy antijamming game. Moreover, Luo et al. in [20] considered a multitier Stackelberg game to model the complex interaction among the nodes.

Regarding secure beamforming in PLS, Lin et al. investigated three different hybrid beamforming architectures to maximize the joint secrecy performance and energy efficiency in satellite–terrestrial integrated networks (STIN) [21]. Furthermore, the authors in [22,23,24] considered joint beamforming and optimization for cooperative STIN transmission. However, since the optimization problem is mathematically intractable, the beamforming strategy needs an iterative algorithm.

In the field of precoding strategy in PLS, Liu et al. employed a source with multiantennas to transmit artificial noise (AN) and information signals as secure precoding in an unmanned aerial vehicle (UAV) network [25]. The authors in [26] sent AN via random and null-space precoders from a massive MIMO base station. Meanwhile, the authors in [27,28] applied cooperative jamming from a multiantenna relay and source to overcome an active eavesdropper. In addition, the authors in [29] used another user in z satellite–terrestrial network as a friendly jammer. However, cooperative jamming requires an additional antenna/node and precoders to transmit the jamming signal aside from the information signal.

In a high-density network, a node selection strategy becomes the common technique to secure the cooperative transmission. The authors in [30,31,32] implemented relay selection to overcome an active eavesdropper, but only in dual-hop transmissions. On the other hand, a multihop network with a larger number of nodes makes the node selection process more complex. Shim et al. studied node selection for a source cluster and relay cluster in [33], while in [34], the authors utilized node selection for a multihop relaying network with power beacons. However, these two studies only studied the secrecy performance under a passive eavesdropper attack, while an active eavesdropper is more destructive to the system performance than a passive eavesdropper.

The authors in [35] proposed a node selection scheme to improve the end-to-end throughput without an eavesdropping attack. The authors in [36] proposed a train-to-train multihop transmission and next relay selection scheme. However, this work did not consider an eavesdropping attack. In [37], the authors exploited the outage performance for short packet communication in a multihop transmission with wireless energy transfer. This work also did not consider an eavesdropping attack. As can be observed, in the multihop transmission context, the improvements of secrecy performance and system throughput are studied. However, the study of a secure multihop transmission under active eavesdropping using a node selection strategy has not been conducted yet.

Different from previous works in [34,35,36,37] that studied various strategies to improve multihop transmission performance without considering an active eavesdropper attack, we propose a cooperative multihop relaying network in confronting a passive and active eavesdropper using node selection schemes. Other works in [30,31,32] only studied the secrecy performance of a dual-hop transmission under active eavesdropping attacks, while with the higher number of node clusters, the end-to-end security is harder to be deployed in multicluster WSNs [10].

## 3. System Model

### 3.1. System Description

Let us consider a multihop transmission in WSN consisting of *K* clusters with *N* nodes in every cluster, as depicted in Figure 1. The desired source (*i*-th node) in the first cluster ($R_{1,i}$) transmits confidential information to a destination (D) through K−1 clusters of relay (R) as the main channel of cooperative transmission. We assume that all nodes in the main channel transmission have a single antenna. Meanwhile, the eavesdropper can overhear the legitimate users’ transmission. If the eavesdropper is operated on the passive mode, the eavesdropper only wiretaps the confidential message since the eavesdropper is equipped with a single antenna. However, when the eavesdropper is operated on the active mode, the eavesdropper wiretaps the confidential message and radiates the jamming signal by using two antennas at the same time. In this paper, we exploit the impact of two eavesdropping “scenarios”, called passive and active. Additionally, we propose node selection “schemes” to improve the system secrecy.

### 3.2. Scenario 1—Passive Eavesdropper

The received signal from the *i*-th node in the *k*-th cluster ($R_{k,i}$) at the *j*-th node in the next cluster ($R_{k+1,j}$) with a passive eavesdropper can be described as
(1)$$y_{k,i,j}^{\mathrm{pas}}=\sqrt{P_{k,i}}h_{k,i,j}x_{k,i}+n_{k+1,j},$$
where $x_{k,i}$ and $P_{k,i}$ denote the transmit signal and power at the *i*-th node in the *k*-th cluster, respectively. $h_{k,i,j}$ denotes the channel coefficient of the $R_{k,i}\to R_{k+1,j}$ link. The channel noise at $R_{k+1,j}$ is denoted by $n_{k+1,j}$ as an additive white Gaussian noise (AWGN) model with zero mean and variance $\sigma_{k+1,j}^{2}$. The signal-to-noise ratio (SNR) of the main channel at the *k*-th hop under a passive eavesdropper attack can be described as
(2)$$\gamma_{k,i,j}^{\mathrm{pas}}=\frac{P_{k,i}{\left|h_{k,i,j}\right|}^{2}}{\sigma_{k+1,j}^{2}}.$$

The received signal at E that only overhears the *k*-th hop data transmission can be expressed as
(3)$$y_{k,i,E}^{\mathrm{pas}}=\sqrt{P_{k,i}}h_{k,i,E}x_{k,i}+n_{E},$$
where $h_{k,i,E}$ indicates the channel coefficient of the $R_{k,i}\to E$ link. $n_{E}$ indicates the channel noise at E with an AWGN model and variance $\sigma_{E}^{2}$. The SNR of the eavesdropper link with a passive mode at the *k*-th hop can be expressed as
(4)$$\gamma_{k,i,E}^{\mathrm{pas}}=\frac{P_{k,i}{\left|h_{k,i,E}\right|}^{2}}{\sigma_{E}^{2}}.$$

### 3.3. Scenario 2—Active Eavesdropper

In the active attack, the received signal at the *k*-th hop data transmission is interfered by the jamming signal from E, which can be described as
(5)$$y_{k,i,j}^{\mathrm{act}}=\underset{\mathrm{information}}{\underset{︸}{\sqrt{P_{k,i}}h_{k,i,j}x_{k,i}}}+\underset{\mathrm{interference}}{\underset{︸}{\sqrt{P_{E}}h_{k,E,j}x_{E}}}+\underset{\mathrm{noise}}{\underset{︸}{n_{k+1,j}}},$$
where $h_{k,E,j}$ denotes the channel coefficient of the $E\to R_{k+1,j}$ link. The jamming signal and power from E are denoted by $x_{E}$ and $P_{E}$, respectively. The signal-to-interference-plus-noise ratio (SINR) of the main channel at the *k*-th hop can be described as
(6)$$\gamma_{k,i,j}^{\mathrm{act}}=\frac{P_{k,i}{\left|h_{k,i,j}\right|}^{2}}{P_{E}{\left|h_{k,E,j}\right|}^{2}+\sigma_{k+1,j}^{2}}.$$

At E, the received signal is affected by the noise of the channel link and the self-interference (SI) from its jamming signal. The received signal at E can be expressed as
(7)$$y_{k,i,E}^{\mathrm{act}}=\underset{\mathrm{information}}{\underset{︸}{\sqrt{P_{k,i}}h_{k,i,E}x_{k,i}}}+\underset{\mathrm{self}-\mathrm{interference}}{\underset{︸}{\sqrt{P_{E}}h_{\mathrm{si}}x_{E}}}+\underset{\mathrm{noise}}{\underset{︸}{n_{E}}},$$
where $h_{\mathrm{si}}$ indicates the channel coefficient of the SI link. After different stages of mitigation, the residual SI (RSI) can be decreased to the noise level [38]. The observation with an RSI component at E can be expressed as
(8)$$y_{k,i,E}^{\mathrm{RSI},\mathrm{act}}=\sqrt{P_{k,i}}h_{k,i,E}x_{k,i}+n_{\mathrm{si}}+n_{E}.$$The SINR of the active eavesdropper at the *k*-th hop transmission can be expressed as
(9)$$\gamma_{k,i,E}^{\mathrm{act}}=\frac{P_{k,i}{\left|h_{k,i,E}\right|}^{2}}{\sigma_{\mathrm{si}}^{2}+\sigma_{E}^{2}}.$$

### 3.4. The Proposed Node Selection Scheme

#### 3.4.1. Random Node Selection (RNS) Scheme

In this scheme, RNS selects a relay node randomly in each cluster without considering the channel information in every node. The SNR of the *k*-th hop main channel transmission with RNS and a passive eavesdropper can be described as
(10)$$\gamma_{k,i^{*},j*}^{\mathrm{RNS},\mathrm{pas}}=\frac{P_{k,i}{\left|h_{k,i,j}\right|}^{2}}{\sigma_{k+1,j}^{2}}.$$Meanwhile, the SNR of the passive eavesdropper link at the *k*-th hop can be written as
(11)$$\gamma_{k,i^{*},E}^{\mathrm{RNS},\mathrm{pas}}=\frac{P_{k,i}{\left|h_{k,i,E}\right|}^{2}}{\sigma_{E}^{2}}.$$

The SINR of the *k*-th hop main channel transmission with RNS and an active eavesdropper can be described as
(12)$$\gamma_{k,i^{*},j*}^{\mathrm{RNS},\mathrm{act}}=\frac{P_{k,i}{\left|h_{k,i,j}\right|}^{2}}{P_{E}{\left|h_{k,E,j}\right|}^{2}+\sigma_{k+1,j}^{2}}.$$Meanwhile, the SINR of the active eavesdropper link at the *k*-th hop can be written as
(13)$$\gamma_{k,i^{*},E}^{\mathrm{RNS},\mathrm{act}}=\frac{P_{k,i}{\left|h_{k,i,E}\right|}^{2}}{\sigma_{\mathrm{si}}^{2}+\sigma_{E}^{2}}.$$We consider this scheme as a benchmark to compare the performance result with the following proposed schemes.

#### 3.4.2. Minimum Node Selection (MNS) Scheme

We propose a minimum selection process to select a relay node in every cluster. The selection criteria of the node with the minimum eavesdropper’s channel gain can be expressed as
(14)$$R_{k,i^{*}}^{\mathrm{MNS}}=\arg\min_{1\leq i\leq N}\left\{|h_{k,i,E}{|}^{2}\right\}.$$The SNR of the *k*-th hop main channel transmission with minimum selection in the presence of a passive eavesdropper can be expressed as
(15)$$\gamma_{k,i^{*},j^{*}}^{\mathrm{MNS},\mathrm{pas}}=\frac{P_{k,i^{*}}{\left|h_{k,i^{*},j^{*}}\right|}^{2}}{\sigma_{k+1,j^{*}}^{2}},$$
where $j^{*}$ denotes the selected node that has already chosen in the next hop. On the other side, the SNR of the passive eavesdropper link at the *k*-th hop becomes
(16)$$\gamma_{k,i^{*},E}^{\mathrm{MNS},\mathrm{pas}}=\frac{P_{k,i}\min_{1\leq i\leq N}\left\{\right|h_{k,i,E}{|}^{2}\}}{\sigma_{E}^{2}}.$$

The SINR of the *k*-th hop main channel transmission with minimum selection and an active eavesdropper can be expressed as
(17)$$\gamma_{k,i^{*},j^{*}}^{\mathrm{MNS},\mathrm{act}}=\frac{P_{k,i^{*}}{\left|h_{k,i^{*},j^{*}}\right|}^{2}}{P_{E}{\left|h_{k,E,j^{*}}\right|}^{2}+\sigma_{k+1,j^{*}}^{2}}.$$Meanwhile, the SINR of the active eavesdropper link at the *k*-th hop becomes
(18)$$\gamma_{k,i^{*},E}^{\mathrm{MNS},\mathrm{act}}=\frac{P_{k,i}\min_{1\leq i\leq N}\left\{\right|h_{k,i,E}{|}^{2}\}}{\sigma_{\mathrm{si}}^{2}+\sigma_{E}^{2}}.$$

#### 3.4.3. Optimal Node Selection (ONS) Scheme

In this selection process, we select the relay node in every cluster that can maximize the secrecy capacity of the system. The main and eavesdropper channels are both considered in an optimal selection process, which can be described by
(19)$$R_{k,i^{*}}^{\mathrm{ONS}}=\arg\max_{1\leq i\leq N}\left\{\log_{2}\left(\frac{1+\gamma_{k,i,j^{*}}}{1+\gamma_{k,i,E}}\right)\right\}.$$The SNR of the *k*-th hop main channel transmission with ONS in the presence of a passive eavesdropper can be written as
(20)$$\gamma_{k,i^{*},j^{*}}^{\mathrm{ONS},\mathrm{pas}}=\frac{P_{k,i}{\left|h_{k,i,j^{*}}\right|}^{2}}{\sigma_{k+1,j^{*}}^{2}}.$$In addition, the SNR of the passive eavesdropper link at the *k*-th hop can be expressed as
(21)$$\gamma_{k,i^{*},E}^{\mathrm{ONS},\mathrm{pas}}=\frac{P_{k,i}{\left|h_{k,i,E}\right|}^{2}}{\sigma_{E}^{2}}.$$

The SINR of the *k*-th hop main channel transmission with ONS in the presence of an active eavesdropper can be written as
(22)$$\gamma_{k,i^{*},j*}^{\mathrm{ONS},\mathrm{act}}=\frac{P_{k,i}{\left|h_{k,i,j^{*}}\right|}^{2}}{P_{E}{\left|h_{k,E,j^{*}}\right|}^{2}+\sigma_{k+1,j^{*}}^{2}}.$$Lastly, the SINR of the active eavesdropper link at the *k*-th hop can be expressed as
(23)$$\gamma_{k,i^{*},E}^{\mathrm{ONS},\mathrm{act}}=\frac{P_{k,i}{\left|h_{k,i,E}\right|}^{2}}{\sigma_{\mathrm{si}}^{2}+\sigma_{E}^{2}}.$$

As can be seen, the SNR models with the proposed scheme and with passive eavesdropping are similar to the well-known selection scheme. However, as can be seen in (Equation 17), (Equation 18), (Equation 22) and (Equation 23), the SINR models with the proposed scheme and with an active eavesdropping attack are different since they have a jamming signal, which cause the derivation complexity that is very challenging. Thus, the proposed node selection schemes still have novel contributions.

## 4. Secrecy Outage Performance Analysis

The system’s secrecy outage probability (SOP) is defined as the probability in which the system secrecy capacity is less than the target secrecy rate ($R_{\mathrm{th}}$), which can be written as
(24)$$P_{\mathrm{SOP}}^{\mathrm{case}}=\mathrm{Pr}\left(\frac{1}{K}\min_{1\leq k\leq K}\log_{2}\left(\frac{1+\gamma_{k,i^{*},j^{*}}^{\mathrm{case}}}{1+\gamma_{k,i^{*},E}^{\mathrm{case}}}\right)<R_{\mathrm{th}}\right),$$
where case∈{c1,c2,c3,c4,c5,c6}. The SOP of the system is associated with the probability that the system cannot securely decode the information [39]. In other words, part of the secret information can be decoded by an eavesdropper. The SOP analysis of the proposed schemes will be presented in six different cases as the combination of the selection scheme and eavesdropper scenario that is shown in Table 1.

We assume that all channels in the system undergo Rayleigh fading, in which the channel gain from X to Y ($|h_{\mathrm{XY}}{|}^{2}$) follows an exponential distribution with mean $\lambda_{\mathrm{XY}}={\left(d_{\mathrm{XY}}/d_{0}\right)}^{-\epsilon }$. $d_{\mathrm{XY}}$ denotes the Euclidean distance between X and Y, while $d_{0}$ represents the reference distance, and ϵ represents the path-loss exponent. For convenience, we define the channel gains as $X_{k,i,j}≜{\left|h_{k,i,j}\right|}^{2}$, $Y_{k,i,E}≜{\left|h_{k,i,E}\right|}^{2}$, and $Z_{k,E,j}≜{\left|h_{k,E,j}\right|}^{2}$. Without loss of generality, we assume $P_{k,i}$ = $P_{R}$ and $\sigma_{k+1,j}^{2}=\sigma_{E}^{2}=\sigma_{\mathrm{si}}^{2}=\sigma^{2}$. We can further suppose $\gamma_{R}$ = $P_{R}/\sigma^{2}$ and $\gamma_{E}$ = $P_{E}/\sigma^{2}$.

### 4.1. Case I: Random Node Selection Scheme under Passive Eavesdropper

From (Equation 24), the SOP with case I can be further written as
(25)$$P_{\mathrm{SOP}}^{c}1=1-\prod_{k=1}^{K}\left[1-\mathrm{Pr}\left(\frac{1+\gamma_{k,i^{*},j^{*}}^{\mathrm{RNS},\mathrm{pas}}}{1+\gamma_{k,i^{*},E}^{\mathrm{RNS},\mathrm{pas}}}<\gamma_{\mathrm{th}}\right)\right],$$
where $\gamma_{\mathrm{th}}=2^{KR_{\mathrm{th}}}$. By relying on the channel characteristic of each link with an RNS scheme, the SOP with case I can be rewritten as
(26)$$\begin{array}{cc} P_{\mathrm{SOP}}^{c1} & =1-\prod_{k=1}^{K}\left[1-\mathrm{Pr}\left(\frac{1+\gamma_{R}X_{k,i,j}}{1+\gamma_{R}Y_{k,i,E}}<\gamma_{\mathrm{th}}\right)\right]\\ \; & =1-\prod_{k=1}^{K}\left[1-\mathrm{Pr}\left(X_{k,i,j}<\frac{\gamma_{\mathrm{th}}-1}{\gamma_{R}}+\gamma_{\mathrm{th}}Y_{k,i,E}\right)\right]. \end{array}$$In order to further calculate $P_{\mathrm{SOP}}^{c1}$, (Equation 26) can be re-expressed as
(27)$$P_{\mathrm{SOP}}^{c1}=1-\prod_{k=1}^{K}\left[1-\underset{\Psi }{\underset{︸}{\int_{0}^{\infty }F_{X_{k,i,j}}\left(\frac{\gamma_{\mathrm{th}}-1}{\gamma_{R}}+\gamma_{\mathrm{th}}y\right)f_{Y_{k,i,E}}\left(y\right)dy}}\right].$$Ψ in (Equation 27) can be rewritten as
(28)$$\begin{array}{cc} \Psi = & \int_{0}^{\infty }\left[1-\exp\left(-\frac{1}{\lambda_{k,i,j}}\left(\frac{\gamma_{\mathrm{th}}-1}{\gamma_{R}}+\gamma_{\mathrm{th}}y\right)\right)\right]\frac{1}{\lambda_{k,i,E}}\exp\left(-\frac{1}{\lambda_{k,i,E}}y\right)dy\\ = & \underset{\Psi_{1a}}{\underset{︸}{\int_{0}^{\infty }\frac{1}{\lambda_{k,i,E}}\exp\left(-\frac{y}{\lambda_{k,i,E}}\right)dy}}-\exp\left(-\frac{\gamma_{\mathrm{th}}-1}{\gamma_{R}\lambda_{k,i,j}}\right)\underset{\Psi_{1b}}{\underset{︸}{\int_{0}^{\infty }\frac{1}{\lambda_{k,i,E}}\exp\left(-\frac{\gamma_{\mathrm{th}}y}{\lambda_{k,i,j}}-\frac{y}{\lambda_{k,i,E}}\right)dy}}. \end{array}$$Relying on the fact [40] (Equation 3.310), i.e., $\int_{0}^{\infty }e^{-px}dx=1/p$, $\Psi_{1a}$ and $\Psi_{1b}$ can be, respectively, re-expressed as
(29)$$\Psi_{1a}=\int_{0}^{\infty }\frac{1}{\lambda_{k,i,E}}\exp\left(-\frac{1}{\lambda_{k,i,E}}y\right)dy=1,$$
(30)$$\Psi_{1b}=\int_{0}^{\infty }\frac{1}{\lambda_{k,i,E}}\exp\left(-\left(\frac{\gamma_{\mathrm{th}}}{\lambda_{k,i,j}}+\frac{1}{\lambda_{k,i,E}}\right)y\right)dy=\frac{\lambda_{k,i,j}}{\gamma_{\mathrm{th}}\lambda_{k,i,E}+\lambda_{k,i,j}}.$$By plugging $\Psi_{1a}$ and $\Psi_{1b}$ into (Equation 28), Ψ can be further expressed as
(31)$$\Psi =1-\frac{\lambda_{k,i,j}}{\gamma_{\mathrm{th}}\lambda_{k,i,E}+\lambda_{k,i,j}}\exp\left(-\frac{\gamma_{\mathrm{th}}-1}{\gamma_{R}\lambda_{k,i,j}}\right).$$By substituting (Equation 31) into (Equation 27) and after some mathematical steps, the closed-form expression for the SOP under case I can be obtained as
(32)$$P_{\mathrm{SOP}}^{c1}=1-\prod_{k=1}^{K}\left[\frac{\lambda_{k,i,j}}{\gamma_{\mathrm{th}}\lambda_{k,i,E}+\lambda_{k,i,j}}\exp\left(-\frac{\gamma_{\mathrm{th}}-1}{\gamma_{R}\lambda_{k,i,j}}\right)\right].$$

### 4.2. Case II: Random Node Selection Scheme under Active Eavesdropper

The SOP under case II can be further written as
(33)$$P_{\mathrm{SOP}}^{c2}=1-\prod_{k=1}^{K}\left[1-\mathrm{Pr}\left(\frac{1+\gamma_{k,i^{*},j^{*}}^{\mathrm{RNS},\mathrm{act}}}{1+\gamma_{k,i^{*},E}^{\mathrm{RNS},\mathrm{act}}}<\gamma_{\mathrm{th}}\right)\right].$$From (Equation 12) and (Equation 13), the SOP of case II can be rewritten as
(34)$$\begin{array}{cc} P_{\mathrm{SOP}}^{c2}= & 1-\prod_{k=1}^{K}\left[1-\mathrm{Pr}\left(\frac{1+\frac{\gamma_{R}X_{k,i,j}}{\gamma_{E}Z_{k,E,j}+1}}{1+\frac{\gamma_{R}Y_{k,i,E}}{2}}<\gamma_{\mathrm{th}}\right)\right]\\ = & 1-\prod_{k=1}^{K}\left[1-\underset{\Phi }{\underset{︸}{\mathrm{Pr}\left(\frac{\gamma_{R}X_{k,i,j}}{\gamma_{E}Z_{k,E,j}+1}<\left(\gamma_{\mathrm{th}}-1\right)+\frac{\gamma_{\mathrm{th}}\gamma_{R}Y_{k,i,E}}{2}\right)}}\right]. \end{array}$$Φ in (Equation 34) can be re-expressed as
(35)$$\begin{array}{cc} \Phi = & \mathrm{Pr}\left(X_{k,i,j}<\frac{\left(\gamma_{\mathrm{th}}-1\right)\left(\gamma_{E}Z_{k,E,j}+1\right)}{\gamma_{R}}+\frac{\gamma_{\mathrm{th}}Y_{k,i,E}\left(\gamma_{E}Z_{k,E,j}+1\right)}{2}\right)\\ = & \int_{0}^{\infty }\underset{\Phi_{1}}{\underset{︸}{\int_{0}^{\infty }F_{X_{k,i,j}}\left(\frac{\left(\gamma_{\mathrm{th}}-1\right)\left(\gamma_{E}z+1\right)}{\gamma_{R}}+\frac{\gamma_{\mathrm{th}}y\left(\gamma_{E}z+1\right)}{2}\right)f_{Y_{k,i,E}}\left(y\right)dy}}f_{Z_{k,E,j}}\left(z\right)dz. \end{array}$$$\Phi_{1}$ in (Equation 35) can be rewritten as
(36)$$\begin{array}{cc} \Phi_{1}= & \int_{0}^{\infty }\left[1-\exp\left(-\frac{\left(\gamma_{\mathrm{th}}-1\right)\left(\gamma_{E}z+1\right)}{\gamma_{R}\lambda_{k,i,j}}-\frac{\gamma_{\mathrm{th}}y\left(\gamma_{E}z+1\right)}{2\lambda_{k,i,j}}\right)\right]\frac{1}{\lambda_{k,i,E}}\exp\left(-\frac{1}{\lambda_{k,i,E}}y\right)dy\\ = & \underset{\Phi_{1a}}{\underset{︸}{\int_{0}^{\infty }\frac{1}{\lambda_{k,i,E}}\exp\left(-\frac{1}{\lambda_{k,i,E}}y\right)dy}}\\ \; & -\exp\left(-\frac{\left(\gamma_{\mathrm{th}}-1\right)\left(\gamma_{E}z+1\right)}{\gamma_{R}\lambda_{k,i,j}}\right)\underset{\Phi_{1b}}{\underset{︸}{\int_{0}^{\infty }\frac{1}{\lambda_{k,i,E}}\exp\left(-\left(\frac{1}{\lambda_{k,i,E}}+\frac{\gamma_{\mathrm{th}}\left(\gamma_{E}z+1\right)}{2\lambda_{k,i,j}}\right)y\right)dy}}. \end{array}$$In order to further calculate $\Phi_{1}$, we rely on the fact [40] (Equation 3.310). $\Phi_{1a}$ and $\Phi_{1b}$ in (Equation 36) can be, respectively, obtained as
(37)$$\Phi_{1a}=\frac{1}{\lambda_{k,i,E}}\int_{0}^{\infty }\exp\left(-\frac{1}{\lambda_{k,i,E}}y\right)dy=\frac{1}{\lambda_{k,i,E}}\lambda_{k,i,E}=1,$$
(38)$$\begin{array}{cc} \Phi_{1b} & =\int_{0}^{\infty }\frac{1}{\lambda_{k,i,E}}\exp\left(-\left(\frac{2\lambda_{k,i,j}+\gamma_{\mathrm{th}}\lambda_{k,i,E}\left(\gamma_{E}z+1\right)}{\lambda_{k,i,E}2\lambda_{k,i,j}}\right)y\right)dy\\ \; & =\frac{2\lambda_{k,i,j}}{2\lambda_{k,i,j}+\gamma_{\mathrm{th}}\lambda_{k,i,E}\left(\gamma_{E}z+1\right)}. \end{array}$$Plugging $\Phi_{1a}$ and $\Phi_{1b}$ into $\Phi_{1}$, (Equation 36) can be rewritten as
(39)$$\Phi_{1}=1-\frac{2\lambda_{k,i,j}}{2\lambda_{k,i,j}+\gamma_{\mathrm{th}}\lambda_{k,i,E}\left(\gamma_{E}z+1\right)}\exp\left(-\frac{\left(\gamma_{\mathrm{th}}-1\right)\left(\gamma_{E}z+1\right)}{\gamma_{R}\lambda_{k,i,j}}\right).$$By substituting $\Phi_{1}$ into (Equation 35) and after some algebraic steps, Φ can be further expressed as
(40)$$\begin{array}{cc} \Phi = & \int_{0}^{\infty }\left[1-\frac{2\lambda_{k,i,j}}{2\lambda_{k,i,j}+\gamma_{\mathrm{th}}\lambda_{k,i,E}\left(\gamma_{E}z+1\right)}\exp\left(-\frac{\left(\gamma_{\mathrm{th}}-1\right)\left(\gamma_{E}z+1\right)}{\gamma_{R}\lambda_{k,i,j}}\right)\right]\\ \; & \times \frac{1}{\lambda_{k,E,j}}\exp\left(-\frac{1}{\lambda_{k,E,j}}z\right)dz\\ = & \underset{\Phi_{2a}}{\underset{︸}{\int_{0}^{\infty }\frac{1}{\lambda_{k,E,j}}\exp\left(-\frac{1}{\lambda_{k,E,j}}z\right)dz}}-\frac{1}{\lambda_{k,E,j}}\exp\left(-\frac{\gamma_{\mathrm{th}}-1}{\gamma_{R}\lambda_{k,i,j}}\right)\\ \; & \times \underset{\Phi_{2b}}{\underset{︸}{\int_{0}^{\infty }\frac{2\lambda_{k,i,j}}{2\lambda_{k,i,j}+\gamma_{\mathrm{th}}\lambda_{k,i,E}+\gamma_{\mathrm{th}}\gamma_{E}\lambda_{k,i,E}z}\exp\left(-\frac{\left(\gamma_{\mathrm{th}}-1\right)\gamma_{E}z}{\gamma_{R}\lambda_{k,i,j}}-\frac{z}{\lambda_{k,E,j}}\right)dz}}. \end{array}$$In order to further express Φ, we rely on the fact [40] (Equation 3.310) and [40] (Equation 3.352.4). $\Phi_{2a}$ and $\Phi_{2b}$ in (Equation 40) can be, respectively, obtained as
(41)$$\Phi_{2a}=\frac{1}{\lambda_{k,E,j}}\int_{0}^{\infty }\exp\left(-\frac{1}{\lambda_{k,E,j}}z\right)dz=\frac{1}{\lambda_{k,E,j}}\lambda_{k,E,j}=1,$$
(42)$$\begin{array}{cc} \Phi_{2b}= & \frac{2\lambda_{k,i,j}}{\gamma_{\mathrm{th}}\gamma_{E}\lambda_{k,i,E}}\int_{0}^{\infty }\frac{1}{\frac{2\lambda_{k,i,j}+\gamma_{\mathrm{th}}\lambda_{k,i,E}}{\gamma_{\mathrm{th}}\gamma_{E}\lambda_{k,i,E}}+z}\exp\left(-\left(\frac{\gamma_{\mathrm{th}}\gamma_{E}\lambda_{k,E,j}-\gamma_{E}\lambda_{k,E,j}+\gamma_{R}\lambda_{k,i,j}}{\gamma_{R}\lambda_{k,i,j}\lambda_{k,E,j}}\right)z\right)dz\\ = & -\frac{2\lambda_{k,i,j}}{\gamma_{\mathrm{th}}\gamma_{E}\lambda_{k,i,E}}\exp\left(\beta_{k}\mu_{k}\right)\mathrm{Ei}\left(-\beta_{k}\mu_{k}\right), \end{array}$$
where $\beta_{k}=\frac{2\lambda_{k,i,j}+\gamma_{\mathrm{th}}\lambda_{k,i,E}}{\gamma_{\mathrm{th}}\gamma_{E}\lambda_{k,i,E}}$, $\mu_{k}=\frac{\gamma_{\mathrm{th}}\gamma_{E}\lambda_{k,E,j}-\gamma_{E}\lambda_{k,E,j}+\gamma_{R}\lambda_{k,i,j}}{\gamma_{R}\lambda_{k,i,j}\lambda_{k,E,j}}$, and Ei(.) mean the exponential integral function [40] (Equation 8.211.1). Again, plugging $\Phi_{2a}$ and $\Phi_{2b}$ into (Equation 40), Φ can be obtained as
(43)$$\Phi =1+\frac{2\lambda_{k,i,j}}{\gamma_{\mathrm{th}}\gamma_{E}\lambda_{k,i,E}\lambda_{k,E,j}}\exp\left(-\frac{\gamma_{\mathrm{th}}-1}{\gamma_{R}\lambda_{k,i,j}}+\beta_{k}\mu_{k}\right)\mathrm{Ei}\left(-\beta_{k}\mu_{k}\right).$$After some algebraic steps, the closed-form expression for the SOP under case II ($P_{\mathrm{SOP}}^{c2}$) can be obtained as
(44)$$P_{\mathrm{SOP}}^{c2}=1-\prod_{k=1}^{K}\left[-\frac{2\lambda_{k,i,j}}{\gamma_{\mathrm{th}}\gamma_{E}\lambda_{k,i,E}\lambda_{k,E,j}}\exp\left(-\frac{\gamma_{\mathrm{th}}-1}{\gamma_{R}\lambda_{k,i,j}}+\beta_{k}\mu_{k}\right)\mathrm{Ei}\left(-\beta_{k}\mu_{k}\right)\right].$$

### 4.3. Case III: Minimal Node Selection with Passive Eavesdropper

The SOP with case III can be further written as
(45)$$P_{\mathrm{SOP}}^{c3}=1-\prod_{k=1}^{K}\left[1-\mathrm{Pr}\left(\frac{1+\gamma_{k,i^{*},j^{*}}^{\mathrm{MNS},\mathrm{pas}}}{1+\gamma_{k,i^{*},E}^{\mathrm{MNS},\mathrm{pas}}}<\gamma_{\mathrm{th}}\right)\right].$$From (Equation 15) and (Equation 16), the SOP under case III can be rewritten as
(46)$$\begin{array}{cc} P_{\mathrm{SOP}}^{c3} & =1-\prod_{k=1}^{K}\left[1-\mathrm{Pr}\left(\frac{1+\gamma_{R}X_{k,i^{*},j^{*}}}{1+\gamma_{R}Y_{k,i^{*},E}}<\gamma_{\mathrm{th}}\right)\right]\\ \; & =1-\prod_{k=1}^{K}\left[1-\underset{\Omega }{\underset{︸}{\mathrm{Pr}\left(X_{k,i^{*},j^{*}}<\frac{\gamma_{\mathrm{th}}-1}{\gamma_{R}}+\gamma_{\mathrm{th}}Y_{k,i^{*},E}\right)}}\right]. \end{array}$$As can be seen, the events of the probability (Equation 46) are not mutually exclusive since it includes $Y_{k,i^{*},E}$. Therefore, by conditioning $Y_{k,i^{*},E}$ = *y*, Ω in (Equation 46) can be re-expressed as
(47)$$\begin{array}{cc} \Omega = & \int_{0}^{\infty }\mathrm{Pr}\left(X_{k,i^{*},j^{*}}<\frac{\gamma_{\mathrm{th}}-1}{\gamma_{R}}+\gamma_{\mathrm{th}}y\right)f_{Y_{k,i^{*},E}}\left(y\right)dy\\ = & \int_{0}^{\infty }\sum_{i=1}^{N}\mathrm{Pr}\left(i=i^{*}\right)\mathrm{Pr}\left(X_{k,i,j^{*}}<\frac{\gamma_{\mathrm{th}}-1}{\gamma_{R}}+\gamma_{\mathrm{th}}y\right)f_{Y_{k,i^{*},E}}\left(y\right)dy. \end{array}$$The following lemmas will help to further calculate $P_{\mathrm{SOP}}^{\mathrm{sc}3}$. First, Lemma 1 helps to obtain the probability of one relay node, which is selected inside a cluster.

**Lemma** **1.**
*The probability of one node selected among N nodes can be expressed as*

(48)
$$\mathrm{Pr}\left(i^{*}=i\right)=\frac{1}{N}.$$



**Proof.** The probability of a node selected based on the criteria in (Equation 14) can be expressed as
(49)$$\begin{array}{cc} \mathrm{Pr}\left(R_{k,i^{*}}=R_{k,i}\right)= & \mathrm{Pr}\left(\min_{m\in N_{k}}\left\{\right|h_{k,m,j}{|}^{2}\}>|h_{k,i,j}{|}^{2}\right)\\ = & \mathrm{Pr}\left(\overset{N}{\underset{m=1,m\neq i}{⋂}}\left(|h_{k,m,j}{|}^{2}>{\left|h_{k,i,j}\right|}^{2}\right)\right). \end{array}$$By conditioning $|h_{k,i,j}{|}^{2}=w$ and assuming that nodes are independent, we can calculate the probability as
(50)$$\begin{array}{cc} \mathrm{Pr}\left(R_{k,i^{*}}=R_{k,i}\right) & =\int_{0}^{\infty }\mathrm{Pr}\left(\overset{N}{\underset{m=1,m\neq i}{⋂}}\left(|h_{k,m,j}{|}^{2}>w\right)\right)f_{|h_{k,i,j}{|}^{2}}\left(w\right)dw\\ \; & =\int_{0}^{\infty }\prod_{m=1,m\neq i}^{N}\left[1-\mathrm{Pr}\left(|h_{k,m,j}{|}^{2}<w\right)\right]f_{|h_{k,i,j}{|}^{2}}\left(w\right)dw\\ \; & =\int_{0}^{\infty }\frac{1}{\lambda_{k,i,j}}\exp\left(-\frac{Nw}{\lambda_{k,i,j}}\right)dw. \end{array}$$Using [40] (Equation 3.310), we can obtain the probability of a node selected as in (Equation 48). The proof of Lemma 1 is concluded. □

The statistical characteristic of the channel gain from the selected node to the next hop will be presented in the following lemma.

**Lemma** **2.**
*Given the selected node $R_{k,i^{*}}$, the CDF and pdf of $|h_{k,i,j^{*}}{|}^{2}$ can be, respectively, expressed as*

(51)
$$F_{|h_{k,i,j^{*}}{|}^{2}}\left(x\right)=1-\exp\left(-\frac{x}{\lambda_{k,i,j}}\right),$$


(52)
$$f_{|h_{k,i,j^{*}}{|}^{2}}\left(x\right)=\frac{1}{\lambda_{k,i,j}}\exp\left(-\frac{x}{\lambda_{k,i,j}}\right).$$



**Proof.** Using the total probability theory, the CDF of $|h_{k,i,j^{*}}{|}^{2}$ can be written as
(53)$$F_{|h_{k,i,j^{*}}{|}^{2}}\left(x\right)=\sum_{i=1}^{N}\mathrm{Pr}\left(R_{k,i^{*}}=R_{k,i}\right)\mathrm{Pr}\left(|h_{k,i,j}{|}^{2}<x\right).$$By relying on (Equation 48) in Lemma 1, (Equation 53) can be further expressed as
(54)$$\begin{array}{cc} F_{|h_{k,i,j^{*}}{|}^{2}}\left(x\right) & =\sum_{i=1}^{N}\frac{1}{N}\mathrm{Pr}\left(|h_{k,i,j}{|}^{2}<x\right)\\ \; & =1-\exp\left(-\frac{x}{\lambda_{k,i,j}}\right). \end{array}$$After some mathematical steps, the pdf of $|h_{k,i,j^{*}}{|}^{2}$ can be obtained as in (Equation 52). The proof of Lemma 2 is concluded. □

Since the MNS scheme at every hop selects the node in a cluster that minimizes the eavesdropper’s channel gain, the statistical characteristic of $|h_{k,i^{*},E}{|}^{2}$ will be presented in the following lemma.

**Lemma** **3.**
*Let $|h_{k,i^{*},E}{|}^{2}=\min_{1\leq i\leq N}{\left|h_{k,i,E}\right|}^{2}$; the CDF and pdf of $|h_{k,i^{*},E}{|}^{2}$ can be, respectively, expressed as*

(55)
$$F_{|h_{k,i^{*},E}{|}^{2}}\left(y\right)=1-\exp\left(-\frac{N}{\lambda_{k,i,E}}y\right),$$


(56)
$$f_{|h_{k,i^{*},E}{|}^{2}}\left(y\right)=\frac{N}{\lambda_{k,i,E}}\exp\left(-\frac{N}{\lambda_{k,i,E}}y\right).$$



**Proof.** From the criteria in (Equation 14), the CDF of $|h_{k,i^{*},E}{|}^{2}$ can be written as
(57)$$\begin{array}{cc} F_{|h_{k,i^{*},E}{|}^{2}}\left(y\right) & =\mathrm{Pr}\left(\min_{1\leq i\leq N}\left\{\right|h_{k,i,E}{|}^{2}\}<y\right)\\ \; & =1-\prod_{i=1}^{N}\left[1-\mathrm{Pr}\left(|h_{k,i,E}{|}^{2}<y\right)\right]. \end{array}$$Ref. (Equation 57) can be further calculated as
(58)$$\begin{array}{cc} F_{|h_{k,i^{*},E}{|}^{2}}\left(y\right)= & 1-\prod_{i=1}^{N}\left[1-\left(1-\exp\left(-\frac{y}{\lambda_{k,i,E}}\right)\right)\right]\\ = & 1-\exp\left(-\frac{N}{\lambda_{k,i,E}}y\right). \end{array}$$After some algebraic steps, the pdf of $|h_{k,i^{*},E}{|}^{2}$ can be obtained as in (Equation 56). The proof of Lemma 3 is concluded. □

By utilizing (Equation 48), (Equation 51), and (Equation 55), Ω in (Equation 47) can be rewritten as
(59)$$\begin{array}{cc} \Omega = & \int_{0}^{\infty }\left[1-\exp\left(-\frac{1}{\lambda_{k,i,j}}\left(\frac{\gamma_{\mathrm{th}}-1}{\gamma_{R}}+\gamma_{\mathrm{th}}y\right)\right)\right]\frac{N}{\lambda_{k,i,E}}\exp\left(-\frac{N}{\lambda_{k,i,E}}y\right)dy\\ = & \underset{\Omega_{1a}}{\underset{︸}{\int_{0}^{\infty }\frac{N}{\lambda_{k,i,E}}\exp\left(-\frac{N}{\lambda_{k,i,E}}y\right)dy}}\\ \; & -\exp\left(-\frac{\gamma_{\mathrm{th}}-1}{\gamma_{R}\lambda_{k,i,j}}\right)\underset{\Omega_{1b}}{\underset{︸}{\int_{0}^{\infty }\frac{N}{\lambda_{k,i,E}}\exp\left(-\left(\frac{\gamma_{\mathrm{th}}}{\lambda_{k,i,j}}+\frac{N}{\lambda_{k,i,E}}\right)y\right)dy}}. \end{array}$$Using [40] (Equation 3.310), $\Omega_{1a}$ and $\Omega_{1b}$ can be further expressed as
(60)$$\Omega_{1a}=\int_{0}^{\infty }\frac{N}{\lambda_{k,i,E}}\exp\left(-\frac{N}{\lambda_{k,i,E}}y\right)dy=1,$$
(61)$$\Omega_{1b}=\int_{0}^{\infty }\frac{N}{\lambda_{k,i,E}}\exp\left(-\left(\frac{\gamma_{\mathrm{th}}\lambda_{k,i,E}+N\lambda_{k,i,j}}{\lambda_{k,i,j}\lambda_{k,i,E}}\right)y\right)dy=\frac{N\lambda_{k,i,j}}{\gamma_{\mathrm{th}}\lambda_{k,i,E}+N\lambda_{k,i,j}}.$$After plugging (Equation 60) and (Equation 61) into (Equation 59), Ω in (Equation 59) can be further expressed as
(62)$$\Omega =1-\frac{N\lambda_{k,i,j}}{\gamma_{\mathrm{th}}\lambda_{k,i,E}+N\lambda_{k,i,j}}\exp\left(-\frac{\gamma_{\mathrm{th}}-1}{\gamma_{R}\lambda_{k,i,j}}\right).$$By substituting Ω into (Equation 46) and after some mathematical calculation steps, the SOP with case III can be obtained as
(63)$$P_{\mathrm{SOP}}^{c3}=1-\prod_{k=1}^{K}\left[\frac{N\lambda_{k,i,j}}{\gamma_{\mathrm{th}}\lambda_{k,i,E}+N\lambda_{k,i,j}}\exp\left(-\frac{\gamma_{\mathrm{th}}-1}{\gamma_{R}\lambda_{k,i,j}}\right)\right].$$

### 4.4. Case IV: Minimal Node Selection with Active Eavesdropper

The SOP of the system with minimum selection in the presence of an active eavesdropper can be written as
(64)$$\begin{array}{cc} P_{\mathrm{SOP}}^{c4}= & 1-\prod_{k=1}^{K}\left[1-\mathrm{Pr}\left(\frac{1+\gamma_{k,i^{*},j^{*}}^{\mathrm{MNS},\mathrm{act}}}{1+\gamma_{k,i^{*},E}^{\mathrm{MNS},\mathrm{act}}}<\gamma_{\mathrm{th}}\right)\right]\\ = & 1-\prod_{k=1}^{K}[1-\underset{\Xi }{\underset{︸}{\mathrm{Pr}\left(\frac{\gamma_{R}X_{k,i^{*},j^{*}}}{\gamma_{E}Z_{k,E,j^{*}}+1}<\left(\gamma_{\mathrm{th}}-1\right)+\frac{\gamma_{\mathrm{th}}\gamma_{R}Y_{k,i^{*},E}}{2}\right)}}]. \end{array}$$As can be seen in (Equation 64), the events of the probability (Equation 64) are not mutually exclusive since they include $Y_{k,i^{*},E}$. Thus, by conditioning $Y_{k,i^{*},E}$ = *y*, Ξ in (Equation 64) can be further expressed as
(65)$$\begin{array}{cc} \Xi = & \mathrm{Pr}\left(X_{k,i^{*},j^{*}}<\frac{\left(\gamma_{\mathrm{th}}-1\right)\left(\gamma_{E}Z_{k,E,j^{*}}+1\right)}{\gamma_{R}}+\frac{\gamma_{\mathrm{th}}Y_{k,i^{*},E}\left(\gamma_{E}Z_{k,E,j^{*}}+1\right)}{2}\right)\\ = & \int_{0}^{\infty }\underset{\Xi_{1}}{\underset{︸}{\mathrm{Pr}\left(X_{k,i^{*},j^{*}}<\frac{\left(\gamma_{\mathrm{th}}-1\right)\left(\gamma_{E}Z_{k,E,j^{*}}+1\right)}{\gamma_{R}}+\frac{\gamma_{\mathrm{th}}y\left(\gamma_{E}Z_{k,E,j^{*}}+1\right)}{2}\right)}}f_{Y_{k,i^{*},E}}\left(y\right)dy. \end{array}$$In order to further represent $P_{\mathrm{SOP}}^{c4}$, $\Xi_{1}$ can be expressed as
(66)$$\begin{array}{cc} \Xi_{1}= & \int_{0}^{\infty }F_{X_{k,i^{*},j^{*}}}\left(\frac{\left(\gamma_{\mathrm{th}}-1\right)\gamma_{E}z}{\gamma_{R}}+\frac{\left(\gamma_{\mathrm{th}}-1\right)}{\gamma_{R}}+\frac{\gamma_{\mathrm{th}}\gamma_{E}yz}{2}+\frac{\gamma_{\mathrm{th}}y}{2}\right)f_{Z_{k,E,j^{*}}}\left(z\right)dz\\ = & \underset{\Xi_{1a}}{\underset{︸}{\int_{0}^{\infty }\frac{1}{\lambda_{k,E,j}}\exp\left(-\frac{z}{\lambda_{k,E,j}}\right)dz}}-\exp\left(-\frac{1}{\lambda_{k,i,j}}\left(\frac{\left(\gamma_{\mathrm{th}}-1\right)}{\gamma_{R}}+\frac{\gamma_{\mathrm{th}}y}{2}\right)\right)\\ \; & \times \;\underset{\Xi_{1b}}{\underset{︸}{\int_{0}^{\infty }\frac{1}{\lambda_{k,E,j}}\exp\left(-\left(\frac{\left(\gamma_{\mathrm{th}}-1\right)\gamma_{E}}{\gamma_{R}\lambda_{k,i,j}}+\frac{\gamma_{\mathrm{th}}\gamma_{E}y}{2\lambda_{k,i,j}}+\frac{1}{\lambda_{k,E,j}}\right)z\right)dz}}. \end{array}$$Relying on the fact [40] (Equation 3.310), $\Xi_{1a}$ and $\Xi_{1b}$ in (Equation 66) can be, respectively, obtained as
(67)$$\Xi_{1a}=\int_{0}^{\infty }\frac{1}{\lambda_{k,E,j}}\exp\left(-\frac{z}{\lambda_{k,E,j}}\right)dz=\frac{1}{\lambda_{k,E,j}}\lambda_{k,E,j}=1,$$
(68)$$\begin{array}{cc} \Xi_{1b} & =\int_{0}^{\infty }\frac{1}{\lambda_{k,E,j}}\exp\left(-\left(\frac{2\left(\gamma_{\mathrm{th}}-1\right)\gamma_{E}\lambda_{k,E,j}+2\gamma_{R}\lambda_{k,i,j}+\gamma_{\mathrm{th}}\gamma_{R}\gamma_{E}\lambda_{k,E,j}y}{2\gamma_{R}\lambda_{k,i,j}\lambda_{k,E,j}}\right)z\right)dz\\ \; & =\frac{2\lambda_{k,i,j}}{\gamma_{\mathrm{th}}\gamma_{E}\lambda_{k,E,j}\left(\frac{2\left(\gamma_{\mathrm{th}}-1\right)\gamma_{E}\lambda_{k,E,j}+2\gamma_{R}\lambda_{k,i,j}}{\gamma_{\mathrm{th}}\gamma_{R}\gamma_{E}\lambda_{k,E,j}}+y\right)}=\frac{2\lambda_{k,i,j}}{\gamma_{\mathrm{th}}\gamma_{E}\lambda_{k,E,j}\left(\eta_{k}+y\right)}, \end{array}$$
where $\eta_{k}=\frac{2\left(\gamma_{\mathrm{th}}-1\right)\gamma_{E}\lambda_{k,E,j}+2\gamma_{R}\lambda_{k,i,j}}{\gamma_{\mathrm{th}}\gamma_{R}\gamma_{E}\lambda_{k,E,j}}$. When plugging $\Xi_{1a}$ and $\Xi_{1b}$ into $\Xi_{1}$, (Equation 66) can be rewritten as
(69)$$\Xi_{1}=1-\frac{2\lambda_{k,i,j}}{\gamma_{\mathrm{th}}\gamma_{E}\lambda_{k,E,j}\left(\eta_{k}+y\right)}\exp\left(-\frac{\gamma_{\mathrm{th}}-1}{\gamma_{R}\lambda_{k,i,j}}-\frac{\gamma_{\mathrm{th}}y}{2\lambda_{k,i,j}}\right).$$

By substituting $\Xi_{1}$ into (Equation 65) and after some algebraic steps, Ξ can be re-expressed as
(70)$$\begin{array}{cc} \Xi = & \int_{0}^{\infty }\left[1-\frac{2\lambda_{k,i,j}}{\gamma_{\mathrm{th}}\gamma_{E}\lambda_{k,E,j}\left(\eta_{k}+y\right)}\exp\left(-\frac{\gamma_{\mathrm{th}}-1}{\gamma_{R}\lambda_{k,i,j}}-\frac{\gamma_{\mathrm{th}}y}{2\lambda_{k,i,j}}\right)\right]\frac{N}{\lambda_{k,i,E}}\exp\left(-\frac{N}{\lambda_{k,i,E}}y\right)dy\\ = & \underset{\Xi_{2a}}{\underset{︸}{\int_{0}^{\infty }\frac{N}{\lambda_{k,i,E}}\exp\left(-\frac{N}{\lambda_{k,i,E}}y\right)dy}}\\ \; & -\frac{2N\lambda_{k,i,j}}{\gamma_{\mathrm{th}}\gamma_{E}\lambda_{k,i,E}\lambda_{k,E,j}}\exp\left(-\frac{\gamma_{\mathrm{th}}-1}{\gamma_{R}\lambda_{k,i,j}}\right)\underset{\Xi_{2b}}{\underset{︸}{\int_{0}^{\infty }\frac{1}{\left(\eta_{k}+y\right)}\exp\left(-\left(\frac{\gamma_{\mathrm{th}}}{2\lambda_{k,i,j}}+\frac{N}{\lambda_{k,i,E}}\right)y\right)dy}}. \end{array}$$Using the fact [40] (Equation 3.310), $\Xi_{2a}$ can be obtained as
(71)$$\Xi_{2a}=\int_{0}^{\infty }\frac{N}{\lambda_{k,i,E}}\exp\left(-\frac{N}{\lambda_{k,i,E}}y\right)dy=\frac{N}{\lambda_{k,i,E}}\frac{\lambda_{k,i,E}}{N}=1.$$In order to further expressed $\Xi_{2b}$, we utilize the fact [40] (Equation 3.352.4). $\Xi_{2b}$ can be re-expressed as
(72)$$\Xi_{2b}=\int_{0}^{\infty }\frac{1}{\eta_{k}+y}\exp\left(-\frac{\gamma_{\mathrm{th}}\lambda_{k,i,E}+2N\lambda_{k,i,j}}{2\lambda_{k,i,j}\lambda_{k,i,E}}y\right)dy=-\exp\left(\eta_{k}\nu_{k}\right)\mathrm{Ei}\left(-\eta_{k}\nu_{k}\right),$$
where $\nu_{k}=\frac{\gamma_{\mathrm{th}}\lambda_{k,i,E}+2N\lambda_{k,i,j}}{2\lambda_{k,i,j}\lambda_{k,i,E}}$. After plugging $\Xi_{2a}$ and $\Xi_{2b}$ into (Equation 70), Ξ can be obtained as
(73)$$\Xi =1+\frac{2N\lambda_{k,i,j}}{\gamma_{\mathrm{th}}\gamma_{E}\lambda_{k,i,E}\lambda_{k,E,j}}\exp\left(-\frac{\gamma_{\mathrm{th}}-1}{\gamma_{R}\lambda_{k,i,j}}+\eta_{k}\nu_{k}\right)\mathrm{Ei}\left(-\eta_{k}\nu_{k}\right).$$By inserting Ξ into (Equation 64), we obtain the SOP of case IV as
(74)$$P_{\mathrm{SOP}}^{c4}=1-\prod_{k=1}^{K}\left[-\frac{2N\lambda_{k,i,j}}{\gamma_{\mathrm{th}}\gamma_{E}\lambda_{k,i,E}\lambda_{k,E,j}}\exp\left(-\frac{\gamma_{\mathrm{th}}-1}{\gamma_{R}\lambda_{k,i,j}}+\eta_{k}\nu_{k}\right)\mathrm{Ei}\left(-\eta_{k}\nu_{k}\right)\right].$$

### 4.5. Case V: Optimal Node Selection with Passive Eavesdropper

According to the definition of the SOP in (Equation 24), the SOP under case V can be further written as
(75)$$\begin{array}{cc} P_{\mathrm{SOP}}^{c5}= & 1-\prod_{k=1}^{K}\left[1-\mathrm{Pr}\left(\max_{1\leq i\leq N}\left\{\frac{1+\gamma_{k,i^{*},j^{*}}^{\mathrm{ONS},\mathrm{pas}}}{1+\gamma_{k,i^{*},E}^{\mathrm{ONS},\mathrm{pas}}}\right\}<\gamma_{\mathrm{th}}\right)\right]\\ = & 1-\prod_{k=1}^{K}\left[1-\prod_{i=1}^{N}\left\{\mathrm{Pr}\left(\frac{1+\gamma_{R}X_{k,i,j^{*}}}{1+\gamma_{R}Y_{k,i,E}}<\gamma_{\mathrm{th}}\right)\right\}\right]. \end{array}$$After some algebraic operations, the SOP under case V can be expressed as
(76)$$\begin{array}{cc} P_{\mathrm{SOP}}^{c5}= & 1-\prod_{k=1}^{K}[1-\prod_{i=1}^{N}\underset{\Theta }{\underset{︸}{\left\{\mathrm{Pr}\left(X_{k,i,j}<\frac{\gamma_{\mathrm{th}}-1}{\gamma_{R}}+\gamma_{\mathrm{th}}Y_{k,i,E}\right)\right\}}}]. \end{array}$$Θ in (Equation 76) can be rewritten as
(77)$$\begin{array}{cc} \Theta = & \int_{0}^{\infty }\left[1-\exp\left(-\frac{1}{\lambda_{k,i,j}}\left(\frac{\gamma_{\mathrm{th}}-1}{\gamma_{R}}+\gamma_{\mathrm{th}}y\right)\right)\right]\frac{1}{\lambda_{k,i,E}}\exp\left(-\frac{y}{\lambda_{k,i,E}}\right)dy\\ = & \underset{\Theta_{1a}}{\underset{︸}{\int_{0}^{\infty }\frac{1}{\lambda_{k,i,E}}\exp\left(-\frac{y}{\lambda_{k,i,E}}\right)dy}}\\ \; & -\exp\left(-\frac{\gamma_{\mathrm{th}}-1}{\gamma_{R}\lambda_{k,i,j}}\right)\underset{\Theta_{1b}}{\underset{︸}{\int_{0}^{\infty }\frac{1}{\lambda_{k,i,E}}\exp\left(-\left(\frac{\gamma_{\mathrm{th}}}{\lambda_{k,i,j}}+\frac{1}{\lambda_{k,i,E}}\right)y\right)dy}}. \end{array}$$Relying on the fact [40] (Equation 3.310), $\Theta_{1a}$ and $\Theta_{1b}$ can be, respectively, obtained as
(78)$$\Theta_{1a}=\int_{0}^{\infty }\frac{1}{\lambda_{k,i,E}}\exp\left(-\frac{y}{\lambda_{k,i,E}}\right)dy=1,$$
(79)$$\Theta_{1b}=\int_{0}^{\infty }\frac{1}{\lambda_{k,i,E}}\exp\left(-\left(\frac{\gamma_{\mathrm{th}}}{\lambda_{k,i,j}}+\frac{1}{\lambda_{k,i,E}}\right)y\right)dy=\frac{\lambda_{k,i,j}}{\gamma_{\mathrm{th}}\lambda_{k,i,E}+\lambda_{k,i,j}}.$$By plugging $\Theta_{1a}$ and $\Theta_{1b}$ into (Equation 77), Θ can be further expressed as
(80)$$\Theta =1-\frac{\lambda_{k,i,j}}{\gamma_{\mathrm{th}}\lambda_{k,i,E}+\lambda_{k,i,j}}\exp\left(-\frac{\gamma_{\mathrm{th}}-1}{\gamma_{R}\lambda_{k,i,j}}\right).$$By substituting (Equation 80) into (Equation 76) and using the binomial theorem [40] (Equation 1.111), we can obtain the closed-form expression for the SOP with case V, which can be expressed as
(81)PSOPc5=1−∏k=1K1−∏i=1N1−λk,i,jγthλk,i,E+λk,i,jexp−γth−1γRλk,i,j=1−∏k=1K1−∑m=0NNm(−1)mλk,i,jγthλk,i,E+λk,i,jmexp−mγth−mγRλk,i,j.

### 4.6. Case VI: Optimal Node Selection with Active Eavesdropper

From (Equation 24), the SOP with case 6 can be further written as
(82)$$P_{\mathrm{SOP}}^{c6}=1-\prod_{k=1}^{K}\left[1-\mathrm{Pr}\left(\max_{1\leq i\leq N}\left\{\frac{1+\gamma_{k,i^{*},j^{*}}^{\mathrm{ONS},\mathrm{act}}}{1+\gamma_{k,i^{*},E}^{\mathrm{ONS},\mathrm{act}}}\right\}<\gamma_{\mathrm{th}}\right)\right].$$The SOP in (Equation 82) can be re-expressed as
(83)$$\begin{array}{cc} P_{\mathrm{SOP}}^{c6}= & 1-\prod_{k=1}^{K}[1-\prod_{i=1}^{N}\{\mathrm{Pr}(\frac{1+\frac{\gamma_{R}X_{k,i,j^{*}}}{\gamma_{E}Z_{k,E,j^{*}}+1}}{1+\frac{\gamma_{R}}{2}Y_{k,i,E}}<\gamma_{\mathrm{th}})\}]\\ = & 1-\prod_{k=1}^{K}\left[1-\prod_{i=1}^{N}\left\{\underset{\Delta }{\underset{︸}{\mathrm{Pr}\left(\frac{\gamma_{R}X_{k,i,j^{*}}}{\gamma_{E}Z_{k,E,j^{*}}+1}<\gamma_{\mathrm{th}}-1+\frac{\gamma_{\mathrm{th}}\gamma_{R}}{2}Y_{k,i,E}\right)}}\right\}\right]. \end{array}$$Δ in (Equation 83) can be given by
(84)$$\begin{array}{cc} \Delta = & \mathrm{Pr}\left(X_{k,i,j^{*}}<\frac{\left(\gamma_{\mathrm{th}}-1\right)\left(\gamma_{E}Z_{k,E,j^{*}}+1\right)}{\gamma_{R}}+\frac{\gamma_{\mathrm{th}}Y_{k,i,E}\left(\gamma_{E}Z_{k,E,j^{*}}+1\right)}{2}\right)\\ = & \int_{0}^{\infty }\underset{\Delta_{1}}{\underset{︸}{\mathrm{Pr}\left(X_{k,i,j^{*}}<\frac{\left(\gamma_{\mathrm{th}}-1\right)\left(\gamma_{E}Z_{k,E,j^{*}}+1\right)}{\gamma_{R}}+\frac{\gamma_{\mathrm{th}}y\left(\gamma_{E}Z_{k,E,j^{*}}+1\right)}{2}\right)}}f_{Y_{k,i,E}}\left(y\right)dy. \end{array}$$In order to further calculate (Equation 84), $\Delta_{1}$ can be written as
(85)$$\begin{array}{cc} \Delta_{1}= & \int_{0}^{\infty }F_{X_{k,i,j^{*}}}\left(\frac{\left(\gamma_{\mathrm{th}}-1\right)\gamma_{E}z}{\gamma_{R}}+\frac{\left(\gamma_{\mathrm{th}}-1\right)}{\gamma_{R}}+\frac{\gamma_{\mathrm{th}}\gamma_{E}yz}{2}+\frac{\gamma_{\mathrm{th}}y}{2}\right)f_{Z_{k,E,j^{*}}}\left(z\right)dz\\ = & \underset{\Delta_{1a}}{\underset{︸}{\int_{0}^{\infty }\frac{1}{\lambda_{k,E,j}}\exp\left(-\frac{z}{\lambda_{k,E,j}}\right)dz}}-\exp\left(-\frac{1}{\lambda_{k,i,j}}\left(\frac{\left(\gamma_{\mathrm{th}}-1\right)}{\gamma_{R}}+\frac{\gamma_{\mathrm{th}}y}{2}\right)\right)\\ \; & \times \underset{\Delta_{1b}}{\underset{︸}{\int_{0}^{\infty }\frac{1}{\lambda_{k,E,j}}\exp\left(-\left(\frac{\left(\gamma_{\mathrm{th}}-1\right)\gamma_{E}}{\gamma_{R}\lambda_{k,i,j}}+\frac{\gamma_{\mathrm{th}}\gamma_{E}y}{2\lambda_{k,i,j}}+\frac{1}{\lambda_{k,E,j}}\right)z\right)dz}}. \end{array}$$Relying on the fact [40] (Equation 3.310), $\Delta_{1a}$ and $\Delta_{1b}$ in (Equation 85) can be, respectively, expressed as
(86)$$\Delta_{1a}=\int_{0}^{\infty }\frac{1}{\lambda_{k,E,j}}\exp\left(-\frac{z}{\lambda_{k,E,j}}\right)dz=\frac{1}{\lambda_{k,E,j}}\lambda_{k,E,j}=1,$$
(87)$$\begin{array}{cc} \Delta_{1b} & =\int_{0}^{\infty }\frac{1}{\lambda_{k,E,j}}\exp\left(-\left(\frac{2\left(\gamma_{\mathrm{th}}-1\right)\gamma_{E}\lambda_{k,E,j}+2\gamma_{R}\lambda_{k,i,j}+\gamma_{\mathrm{th}}\gamma_{R}\gamma_{E}\lambda_{k,E,j}y}{2\gamma_{R}\lambda_{k,i,j}\lambda_{k,E,j}}\right)z\right)dz\\ \; & =\frac{2\lambda_{k,i,j}}{\gamma_{\mathrm{th}}\gamma_{E}\lambda_{k,E,j}\left(\frac{2\left(\gamma_{\mathrm{th}}-1\right)\gamma_{E}\lambda_{k,E,j}+2\gamma_{R}\lambda_{k,i,j}}{\gamma_{\mathrm{th}}\gamma_{R}\gamma_{E}\lambda_{k,E,j}}+y\right)}=\frac{2\lambda_{k,i,j}}{\gamma_{\mathrm{th}}\gamma_{E}\lambda_{k,E,j}\left(\eta_{k}+y\right)}, \end{array}$$
where $\eta_{k}$ is defined as (Equation 68). Again, when plugging $\Delta_{1a}$ and $\Delta_{1b}$ into (Equation 85), $\Delta_{1}$ can be rewritten as
(88)$$\Delta_{1}=1-\frac{2\lambda_{k,i,j}}{\gamma_{\mathrm{th}}\gamma_{E}\lambda_{k,E,j}\left(\eta_{k}+y\right)}\exp\left(-\frac{\gamma_{\mathrm{th}}-1}{\gamma_{R}\lambda_{k,i,j}}-\frac{\gamma_{\mathrm{th}}y}{2\lambda_{k,i,j}}\right).$$

By substituting $\Delta_{1}$ into (Equation 84) and after some algebraic steps, Δ can be re-expressed as
(89)$$\begin{array}{cc} \Delta = & \int_{0}^{\infty }\left[1-\frac{2\lambda_{k,i,j}}{\gamma_{\mathrm{th}}\gamma_{E}\lambda_{k,E,j}\left(\eta_{k}+y\right)}\exp\left(-\frac{\gamma_{\mathrm{th}}-1}{\gamma_{R}\lambda_{k,i,j}}-\frac{\gamma_{\mathrm{th}}y}{2\lambda_{k,i,j}}\right)\right]\frac{1}{\lambda_{k,i,E}}\exp\left(-\frac{1}{\lambda_{k,i,E}}y\right)dy\\ = & \underset{\Delta_{2a}}{\underset{︸}{\int_{0}^{\infty }\frac{1}{\lambda_{k,i,E}}\exp\left(-\frac{1}{\lambda_{k,i,E}}y\right)dy}}\\ \; & -\frac{2\lambda_{k,i,j}}{\gamma_{\mathrm{th}}\gamma_{E}\lambda_{k,i,E}\lambda_{k,E,j}}\exp\left(-\frac{\gamma_{\mathrm{th}}-1}{\gamma_{R}\lambda_{k,i,j}}\right)\underset{\Delta_{2b}}{\underset{︸}{\int_{0}^{\infty }\frac{1}{\left(\eta_{k}+y\right)}\exp\left(-\left(\frac{\gamma_{\mathrm{th}}}{2\lambda_{k,i,j}}+\frac{1}{\lambda_{k,i,E}}\right)y\right)dy}}. \end{array}$$Using the fact [40] (Equation 3.310), $\Delta_{2a}$ can be obtained as
(90)$$\Delta_{2a}=\int_{0}^{\infty }\frac{1}{\lambda_{k,i,E}}\exp\left(-\frac{1}{\lambda_{k,i,E}}y\right)dy=\frac{1}{\lambda_{k,i,E}}\lambda_{k,i,E}=1.$$In order to further express $\Delta_{2b}$, we utilize the fact [40] (Equation 3.352.4). $\Delta_{2b}$ can be re-expressed as
(91)$$\Delta_{2b}=\int_{0}^{\infty }\frac{1}{\eta_{k}+y}\exp\left(-\frac{\gamma_{\mathrm{th}}\lambda_{k,i,E}+2\lambda_{k,i,j}}{2\lambda_{k,i,j}\lambda_{k,i,E}}y\right)dy=-\exp\left(\eta_{k}\alpha_{k}\right)\mathrm{Ei}\left(-\eta_{k}\alpha_{k}\right),$$
where $\alpha_{k}=\frac{\gamma_{\mathrm{th}}\lambda_{k,i,E}+2\lambda_{k,i,j}}{2\lambda_{k,i,j}\lambda_{k,i,E}}$. After plugging $\Delta_{2a}$ and $\Delta_{2b}$ into (Equation 89), Δ can be obtained as
(92)$$\Delta =1+\frac{2\lambda_{k,i,j}}{\gamma_{\mathrm{th}}\gamma_{E}\lambda_{k,i,E}\lambda_{k,E,j}}\exp\left(-\frac{\gamma_{\mathrm{th}}-1}{\gamma_{R}\lambda_{k,i,j}}+\eta_{k}\alpha_{k}\right)\mathrm{Ei}\left(-\eta_{k}\alpha_{k}\right).$$By plugging Δ into (Equation 83) and using the binomial theorem [40] (Equation 1.111), we obtain the SOP of case VI as
(93)PSOPc6=1−∏k=1K1−∏i=1N1+2λk,i,jγthγEλk,i,Eλk,E,jexp−γth−1γRλk,i,j+ηkαkEi(−ηkαk)=1−∏k=1K[1−∑m=0NNm2λk,i,jγthγEλk,i,Eλk,E,jmexp−mγth−mγRλk,i,j+mηkαk×Ei(−ηkαk)m].

## 5. Performance Evaluations

In this section, we exploit the impact of the active eavesdropping attack and the proposed node selection schemes on the secrecy performance. Unless otherwise stated, the simulation parameters are presented in Table 2.

Figure 2 shows the effect of $\gamma_{R}$ on the SOP. As can be seen, when the transmitted SNR increases, the SOP is decreased. It can be explained by the SNR of the main channel, which is improved, as well as that of the eavesdropper channel. However, the impact of the SNR of the main channel is more than that of the eavesdropper channel. Additionally, when the eavesdropper generates a jamming signal, i.e., active eavesdropping attack, the SOP is higher than the passive eavesdropping attack. The reason is that the jamming signal can degrade the SINR of the main channel, which leads to reducing the difference between the main channel and eavesdropper channel capacities. In order to counteract the eavesdropper attack, different from the other node selection schemes that select the node randomly or utilize only eavesdropper channel information, the proposed ONS scheme uses both main channel and eavesdropper channel information. Thus, the proposed ONS scheme shows the most robust secrecy performance compared with that of other node selection schemes. The comparison between simulation and analytical results is in good agreement, validating the correctness of our derivation approaches. Hence, the following results only provide the theoretical results.

Figure 3 shows the effect of $\gamma_{E}$ towards the SOP of a cooperative multihop relaying network. As can be seen, the SOP for passive eavesdropper scenarios is constant. It can be explained by the fact that the eavesdropper does not radiate a jamming signal, only overhearing the legitimate users’ transmission. In contrast, the active eavesdropper radiates the jamming signal. Thus, when the jamming SNR increases, the SOP with case II, case IV, and case VI is increased. It means that the difference between main channel and eavesdropper channel capacities is reduced. In Figure 3, ONS has the lowest SOP among all the node selection schemes because ONS selects a node in every cluster that gives the maximum secrecy rate. Therefore, the probability of the system secrecy being outage in an ONS scheme becomes minimum. On the contrary, RNS selects a node randomly, and MNS selects a node only based on the eavesdropper link that makes low system secrecy capacity and high SOP.

Figure 4 shows the effect of $R_{\mathrm{th}}$ on the SOP. As shown in Figure 4, with the higher target secrecy data rate, the SOP is increased. The reason is that a higher target secrecy rate correlates with a higher threshold level of the system secrecy being outage. Therefore, the probability of the outage event becomes higher. In Figure 4, a passive eavesdropper scenario produces a better SOP than an active scenario because the system secrecy in a passive attack is higher; then the probability of an outage event is lower. Once again, ONS has the most robust SOP between all node selection schemes in Figure 4 because ONS utilizes both main channel and eavesdropper channel information to select a node. Case V with an ONS scheme and passive eavesdropper scenario has the lowest SOP among all cases. One of the reasons is that ONS selects the best node in every cluster that maximizes the secrecy capacity rate. Furthermore, the passive mode of an eavesdropper allows a less significant attack to the main channel capacity.

Figure 5 illustrates the relation between the SOP and *N*. As can be seen, the number of nodes in a cluster does not have an impact on the SOP of an RNS scheme. An RNS scheme’s SOP in cases I and II is constant for all *N* values because RNS only selects one node randomly regardless of many nodes that can be chosen. ONS and MNS schemes, on the other side, have a lower SOP when the number of nodes per cluster increases because there are more nodes that can be selected to relay the confidential information. In a cluster with more nodes, ONS and MNS schemes have more possibility to choose a node that has better secrecy capacity; then the probability of the secrecy being outage declines. Additionally, an active eavesdropper scenario yields a higher SOP due to more attacks on the main channel capacity than a passive eavesdropper scenario. An active eavesdropper radiates a jamming signal aiming to harm the main channel transmission. Hence, an active eavesdropper is more destructive.

Figure 6 illustrates the impact of *K* on the SOPs. The SOP is decreased when the number of hops increases because there are more hops that can be chosen as the minimum system secrecy capacity rate. This corresponds to (Equation 24), where the SOP is inversely proportional to the number of hops. As can be observed more from Figure 6, a passive eavesdropper scenario produces a lower SOP due to its harmless attack on the main channel capacity than an active mode. The passive eavesdropping scenarios do not radiate a jamming signal, different from the active eavesdropping scenario. In Figure 6, an ONS scheme has the lowest SOP among all the node selection schemes since ONS selects the best node in every cluster that gives the maximum secrecy capacity based on the main channel and eavesdropper channel information. On the other hand, RNS selects a node randomly and MNS selects a node only based on the minimum eavesdropper channel gain that cannot increase the secrecy capacity significantly.

In order to further analyze the secrecy performance of the considered multihop transmission system, we calculate the system secrecy throughput, which is mathematically defined as [38]
(94)$$T_{\mathrm{case}}=\left(1-P_{\mathrm{SOP}}^{\mathrm{case}}\right)R_{\mathrm{th}}.$$

Secrecy throughput is defined as the achievable secrecy rate of the system [41]. Figure 7 presents the system’s secrecy throughput as a function of $\gamma_{R}$. The increment of the node transmit SNR results in the increase in the system’s secrecy throughput. High SNR in the relay nodes increases the main channel capacity higher than the eavesdropper channel capacity because of the eavesdropping counteracting at the node selection process in every cluster. This eventually increases the overall system secrecy rate. The ONS scheme in Figure 7 has the highest secrecy throughput since ONS selects a node with the maximum secrecy capacity in every cluster. On the other hand, an RNS scheme has the lowest secrecy throughput because RNS selects a node randomly without considering the main channel and eavesdropper channel information. Active eavesdropping scenarios bring more destruction in the system secrecy throughput than passive eavesdropping scenarios due to the transmitted jamming signal that degrades the main channel capacity rate.

Figure 8 presents the system’s secrecy throughput as a function of $\gamma_{E}$. As can be seen, a passive scenario of an eavesdropper in cases I, III, and V has a constant secrecy throughput because a passive eavesdropper only overhears the main channel information without transmitting any jamming signal. Cases II, IV, and VI under an active eavesdropper attack has a declination of secrecy throughput as a jamming SNR is increasing, since more jamming is interfering the main channel transmission, and eventually, the difference between main channel and eavesdropper channel capacities is decreasing. Case II with an RNS scheme has the lowest secrecy throughput because a node in every cluster is chosen randomly regardless of its channel information. However, the utilization of an ONS scheme can increase the secrecy throughput higher than RNS and MNS, since ONS selects the best node with the maximum secrecy capacity based on both main channel and eavesdropper channel information.

Finally, we turn our attention to the complexity order. The complexity order is defined as the amount of channel information to select the node and transmit a signal. Table 3 presents the complexity order for every case of a system. Since an RNS scheme selects the node randomly, the RNS scheme does not utilize the channel information at the node selection step. Thus, the RNS scheme shows the lowest complexity order among the considered schemes. An MNS scheme only utilizes the eavesdropper channel information to select the node in each cluster, so the complexity for the selection process grows as N×K. An ONS scheme utilizes both main channel and eavesdropper channel information to select a node in every cluster. Thus, an ONS scheme requires the most channel information among the considered schemes.

Figure 9 visualizes the trade-off between the SOP and complexity as a function of *K*. As can be seen, when the number of hops increases, the complexity order of the proposed node selection schemes is increased, while the SOP is decreased. More specifically, the RNS scheme’s complexity order increases linearly, while the SOP is decreased. On the other hand, the ONS scheme’s complexity order increases when the number of hops increases, while the SOP is decreased significantly. Finally, the complexity order of the MNS scheme is slightly increased, while the SOP of the MNS scheme is decreased and still providing secure transmission. From these phenomena, complexity order and SOP have a trade-off. However, though the complexity order of an ONS scheme is increased, the secrecy performance is significantly improved. Thus, we can conclude that the ONS scheme has advantage against complexity order.

## 6. Conclusions

This paper studied the impact of the eavesdropping attack on the multihop transmission system for sensor networks. More specifically, we exploited the active and passive eavesdropping attacks. The active eavesdropping attack can overhear the legitimate users’ transmission and radiate the jamming signal to degrade the main channel condition. The passive eavesdropping attack only overhears the legitimate users’ transmission. As a counteraction, in order to protect the confidential message against various eavesdropping attacks, we proposed the node selection schemes called MNS scheme and ONS scheme. Since the MNS scheme only required the eavesdropper channel information to select the node in each cluster, it had low complexity and slightly improved the secrecy performance. Meanwhile, the ONS scheme selected the node in each cluster to maximize the secrecy capacity. Thus, the ONS scheme showed high complexity and significantly improved the secrecy performance. We derived the closed-form expression for the SOP with a different eavesdropping attack and node selection scheme. Numerical results showed that an active eavesdropping attack is more destructive compared with a passive attack since an active eavesdropper generated the jamming signal. The ONS scheme utilized both the main channel and eavesdropper channel to select the node in each cluster, which brought the most robust secrecy performance compared with the other node selection schemes. Additionally, through various numerical results, the proposed node selection scheme and different eavesdropping attack on the secrecy performance were discussed. In order to expand this work, we try to develop the secure routing protocol that utilizes the physical layer security concept and blockchain to protect a confidential packet against a sniffing attack.

## Figures and Tables

**Figure 1 sensors-23-07653-f001:**
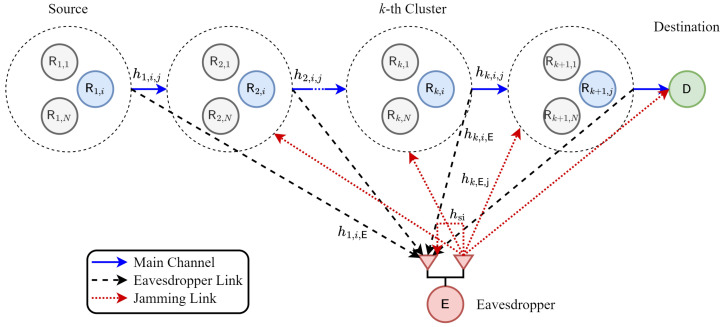
The proposed system model of the multihop transmission.

**Figure 2 sensors-23-07653-f002:**
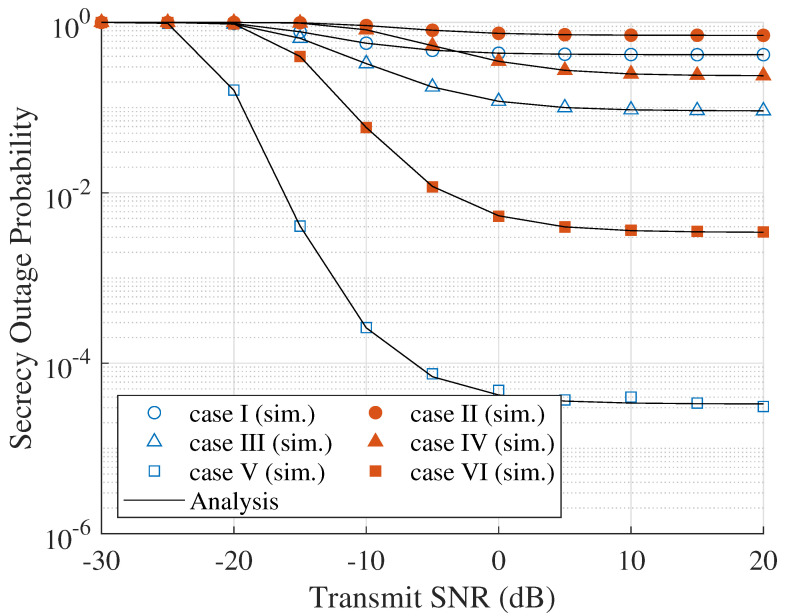
SOP versus node transmit SNR ($\gamma_{R}$).

**Figure 3 sensors-23-07653-f003:**
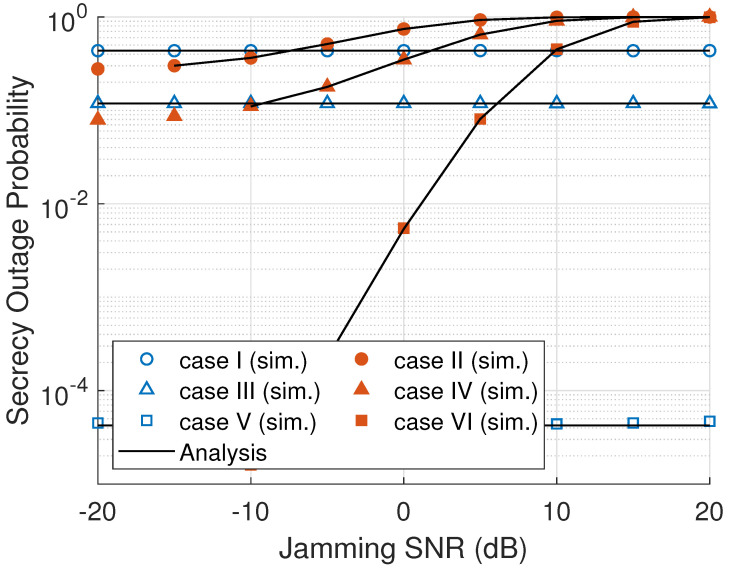
SOP versus eavesdropper’s jamming SNR ($\gamma_{E}$).

**Figure 4 sensors-23-07653-f004:**
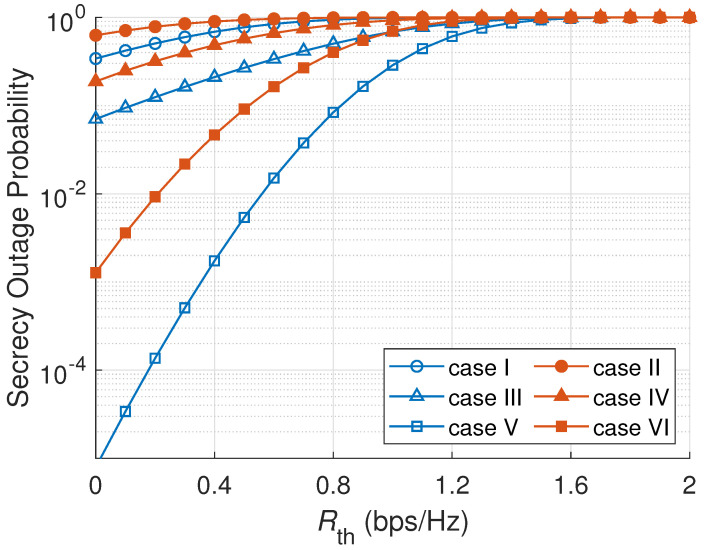
SOP versus target secrecy data rate ($R_{\mathrm{th}}$).

**Figure 5 sensors-23-07653-f005:**
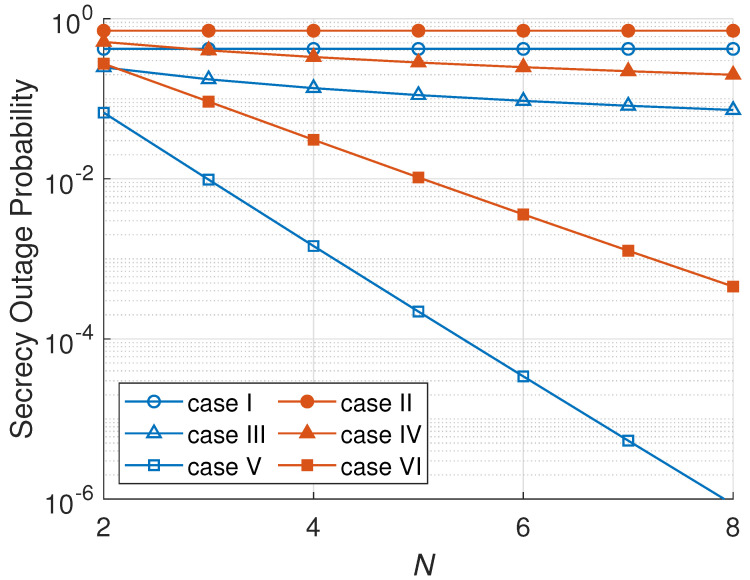
SOP versus number of nodes per cluster (*N*).

**Figure 6 sensors-23-07653-f006:**
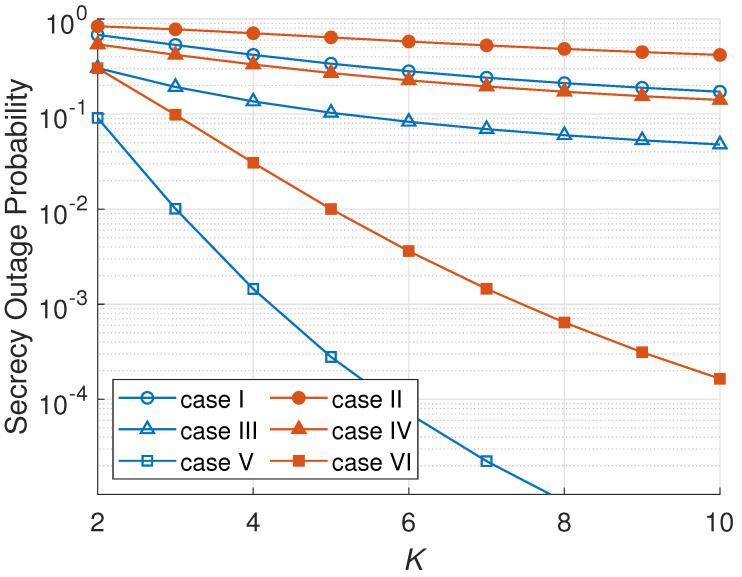
SOP versus number of hops (*K*).

**Figure 7 sensors-23-07653-f007:**
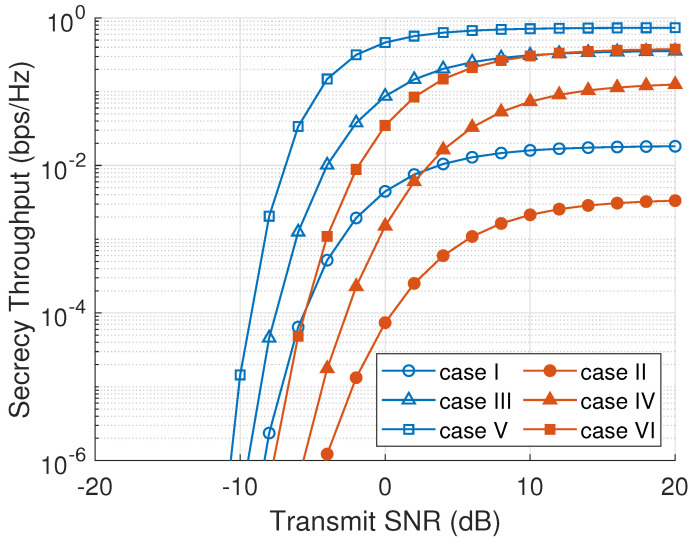
Secrecy throughput versus node transmit SNR ($\gamma_{R}$).

**Figure 8 sensors-23-07653-f008:**
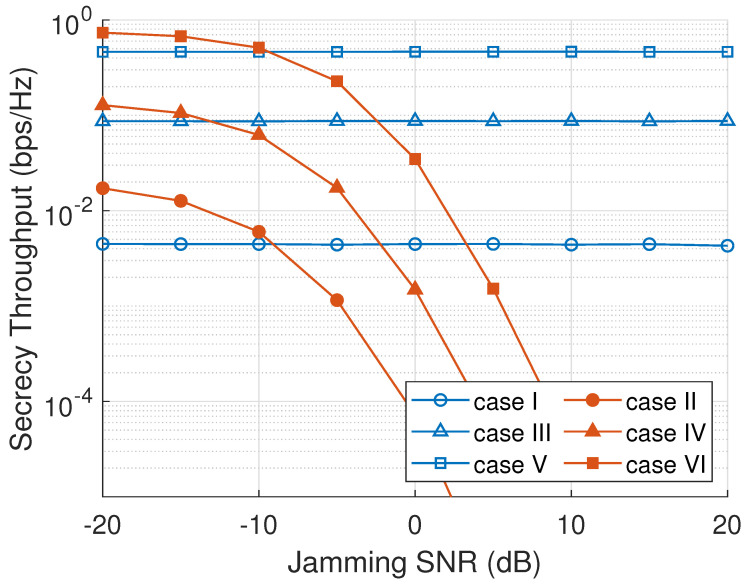
Secrecy throughput versus eavesdropper jamming SNR ($\gamma_{E}$).

**Figure 9 sensors-23-07653-f009:**
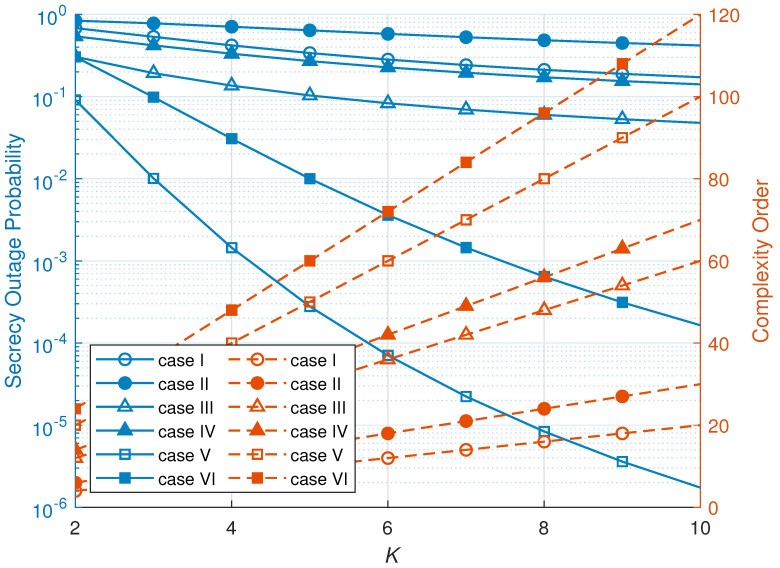
Complexity versus number of hops (*K*).

**Table 1 sensors-23-07653-t001:** Selection and scenario combinations.

Case	Node Selection Scheme	Eavesdropper Scenario
Case I (c1)	RNS	passive
Case II (c2)	RNS	active
Case III (c3)	MNS	passive
Case IV (c4)	MNS	active
Case V (c5)	ONS	passive
Case VI (c6)	ONS	active

**Table 2 sensors-23-07653-t002:** Simulation parameters.

Parameters	Value
Distance between S and D ($d_{\mathrm{SD}}$)	10 m
Position of S	(0, 0)
Position of D	(10, 0)
Position of E	(5, −5)
Position of $R_{k}$	($d_{\mathrm{SD}}k/K$, 0)
Number of hops (*K*)	4 hops
Number of nodes (*N*) in each cluster	6 nodes
Reference distance ($d_{0}$)	10 m
Path-loss exponent (ϵ)	2.7
Target secrecy rate ($R_{\mathrm{th}}$)	0.1 bps/Hz
Node transmit SNR ($\gamma_{R}$)	10 dB
Eavesdropper jamming SNR ($\gamma_{E}$)	0 dB

**Table 3 sensors-23-07653-t003:** Complexity order of the schemes.

Case	I	II	III	IV	V	VI
Scheme	RNS	RNS	MNS	MNS	ONS	ONS
Attack	passive	active	passive	active	passive	active
Complexity	2K	3K	(N+2)K	(N+3)K	(2N+2)K	(2N+4)K

## Data Availability

Not applicable.

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
