# Peer review of "Secrecy Performance Analysis of Cooperative Multihop Transmission for WSNs under Eavesdropping Attacks†"

_sensors, 2023, doi:10.3390/s23177653_

Round 1

Reviewer 1 Report

Please see the attached comments.

Minor editing of English language is needed.

Reviewer 2 Report

In this paper, the authors explore eavesdropping vulnerabilities in wireless sensor networks. They introduced the Opportunistic Node Selection (ONS) scheme, to optimizes selection of nodes to minimize eavesdropping risks. They show the efficiency of ONS against both passive and active eavesdroppers, through an analysis of secrecy performance metrics, even though ONS comes with higher complexity.

I would give the following suggestion to the authors:

Expand on the real-world implications and applications of the research. Why is eavesdropping a concern in sensor networks? In which scenarios or industries is this most relevant? Given that the ONS scheme has higher complexity, it would be beneficial to provide a clearer breakdown or visualization of the trade-offs. For instance, how does the increased security weigh against the increased computational burden in real-world scenarios?

 I would also suggest the authors to clearly outline avenues for future research. This could be in the form of expanding the network size, considering different types of eavesdropping attacks, or integrating with other security mechanisms.

There are some inconsistencies with the references. Some references for conference papers include the phrase "In Proceedings of", while others just mention the conference. 

Reviewer 3 Report

In this paper, the authors aim to address the security problems of wireless sensor networks. The topic selected in this work is both interesting and of significant importance. While the academic writing meets expectations, the technical writing requires further refinement. During the review process, I have identified several points, which are detailed in the 'Points to the Authors' section.

·        The author is advised to manage the contribution points in the introduction section. Generally, they are mentioned in this part. So, this will improve the structure of the paper.

·        Why this work is important in presence of “DOI: 10.1109/ACCESS.2020.2978303”,  and doi.org/10.1016/j.adhoc.2016.10.007. In the revised paper, add these references and claim your contribution.

·        In figure 1, some of nodes are without their description. Please check for this, and explain in the paper, this will increase the reader interest in this work

·        Some of the notations are used without their description. At first use, define every symbol and notation.

·        What are the other possible applications where the proposed model can be implemented. If any is there update it in the paper.

English is fine and understandable 

Round 2

Reviewer 1 Report

The authors have well addressed all my concerns, no further comments.

Reviewer 3 Report

Thanks for addressing my previous round comments. At this point, I do not have further suggestion for the authors